# Endocrine-Disrupting Chemicals and the Effects of Distorted Epigenetics on Preeclampsia: A Systematic Review

**DOI:** 10.3390/cells14070493

**Published:** 2025-03-26

**Authors:** Balu Usha Rani, Ramasamy Vasantharekha, Winkins Santosh, Thangavelu Swarnalingam, Seetharaman Barathi

**Affiliations:** 1Endocrine Disruption and Reproductive Toxicology Laboratory (EDART), Department of Biotechnology, School of Bioengineering, SRM Institute of Science and Technology, Kattankulathur 603203, India; ub3887@srmist.edu.in (B.U.R.); vasanthr@srmist.edu.in (R.V.); 2Toxicology Research on Endocrine Disruptors (TRENDS) Laboratory, PG & Research Department of Advanced Zoology and Biotechnology, Government Arts College, Nandanam, Chennai 600035, India; santoshmcc@yahoo.com; 3Department of Critical Care Medicine, SRM Medical College Hospital & Research Centre, SRM Institute of Science and Technology, Kattankulathur 603203, India; swarnalt@srmist.edu.in

**Keywords:** preeclampsia, placenta, endocrine-disrupting chemicals, phthalates, phenols, epigenomics, microRNAs, DNA methylation

## Abstract

Background: Preeclampsia (PE) is a critical complication of pregnancy that affects 3% to 5% of all pregnancies and has been linked to aberrant placentation, causing severe maternal and fetal illness and death. Objectives: This systematic review aims to elucidate the association of in-utero endocrine-disrupting chemical (EDC) exposure and microRNAs and their imprinted genes from prenatal and maternal circulation of PE patients. Methods: Databases such as PubMed, PubMed Central, ScienceDirect, the Comparative Toxicogenomics Database (CTD), ProQuest, EBSCOhost, and Google Scholar were utilized to search for articles that investigate the relationships between selected EDCs and epigenetic events such as DNA methylation and microRNAs that are associated with PE. Results: A total of 29 studies were included in the database search. Altered expression of microRNAs (miR-15a-5p, miR-142-3p, and miR-185) in the placenta of PE patients was positively associated with the urinary concentration of phthalates and phenols in the development of the disease in the first trimester. EDCs such as phenols, phthalates, perfluoroalkyl substances (PFOAs), polybrominated diphenyl ethers (PBDEs), and organochlorine phosphates (OCPs) have been reported to be associated with hypertensive disorders in pregnancy. miRNA-31, miRNA-144, miRNA-145, miRNA-210, placental specific clusters (C14MC, and C19MC) may be used as possible targets for PE because of their potential roles in the onset and progression of PE. Conclusions: Prenatal EDC exposure, including exposure to BPA, showed association with signaling pathways including estrogen, sFlt-1/PlGF, ErbB, MAPK/ERK, and cholesterol mechanisms with placental hemodynamics. Even low EDC exposures leave altered epigenetic marks throughout gestation, which might cause PE complications.

## 1. Introduction

Preeclampsia (PE) is recognized as a new-onset gestational hypertension with or without proteinuria and/or end-organ damage. It often leads to severe outcomes, including intrauterine growth restriction (IUGR), preterm birth (PTM), and, in extreme cases, maternal and fetal mortality. It affects 3–5% of all pregnancies, and it is found to be a primary cause of maternal and perinatal illness and death [1]. Despite decades of research, the precise etiology of PE remains elusive, although it is widely acknowledged to be influenced by a complex interplay of genetic, environmental, and epigenetic factors. Recent studies highlight the potential role of endocrine-disrupting chemicals (EDCs) in exacerbating the risk of PE through their profound effects on the epigenome. PE and other pregnancy problems, particularly those caused by EDCs, have been linked to increased hypertension [2]. EDCs encompass a diverse group of substances commonly present in household items and food products, including phthalates, bisphenol A (BPA), PBDEs, and PFAS compounds such as perfluoro nonanoic acid (PFNA), perfluorooctanoic acid (PFOA), perfluoro octane sulfonate (PFOS), and perfluoro hexane sulfonate (PFHxS), and they contribute significantly to the world’s illness burden [3,4]. A questionnaire survey identified links of exposure pathways of these EDCs, including through cosmetics, processed foods, dairy products, and non-stick utensils, to various biochemical alterations in PE [5]. About 800 such compounds have been identified as disrupting several endocrine pathways, sometimes mimicking endocrine hormones [6]. These EDCs can mimic normal endocrine systems or hormones such as estrogen, androgen, testosterone, or thyroid hormones [7,8]. As a central mediator of maternal–fetal exchange, the placenta is a key target of these chemicals. Further, it might lead to endothelial dysfunction in the placenta.

EDCs can disrupt the placenta’s structure and function by altering epigenetic regulatory mechanisms, including DNA methylation, histone modifications, and non-coding RNA expression. Epigenetics refers to heritable changes in gene expression that do not involve alterations to the underlying DNA sequence. These changes are crucial for orchestrating normal placental development, trophoblast differentiation, and the maternal immune response [9]. In PE, epigenetic dysregulation is driven by both intrinsic and extrinsic factors, which have been implicated in the abnormal remodeling of spiral arteries, oxidative stress, and inflammation [3]. EDCs interfere with epigenetic programming and have emerged as significant contributors to the onset and progression of PE.

The most compelling epigenetic regulators in PE are microRNAs (miRNAs), small non-coding RNAs that post-transcriptionally regulate gene expression by binding to messenger RNAs (mRNAs) [10]. miRNAs are crucial for maintaining placental homeostasis, including processes such as trophoblast invasion, angiogenesis, and immune modulation. Dysregulation of miRNA expression, often driven by EDC exposure, has been linked to aberrant placental development in PE. Evidence showed altered expression of miR-210, miR-155, and the miR-17-92 cluster in the placenta and maternal circulation of women with PE. These miRNAs play a crucial role in regulating key pathways involved in the pathophysiology of PE, including hypoxia-inducible factors (HIFs), angiogenic balance (VEGF and sFlt-1), and immune signaling, collectively contributing to disease progression. Apart from maternal circulation, tissue-specific miRNAs are also highly expressed in the placenta. Placental miRNA clusters in the human genome consist of C14MC and C19MC, each of which contains 46 or more unique miRNAs. Evidence suggests that C19MC clusters increase from early to late pregnancy, while C14MC clusters decline during this gestational period in PE [11,12].

Pathway analysis revealed that EDC-induced miRNA dysregulation can disrupt several biological networks essential for pregnancy. For instance, miR-210, often termed the “master hypoxamiR”, was reported to be upregulated in hypoxic conditions common in PE and to suppress mitochondrial function and trophoblast invasion by targeting iron–sulfur cluster assembly enzyme (ISCU) [13]. Similarly, miR-155 is known to modulate inflammation and immune tolerance by targeting cytokine signaling pathways, including the suppression of SOCS1 (suppressor of cytokine signaling 1), contributing to systemic inflammation in PE. EDC exposure has been shown to influence DNA methylation patterns at miRNA promoter regions, further amplifying their dysregulation.

The critical pathway involves the peroxisome proliferator-activated receptors (PPARs), which are nuclear receptors regulating lipid metabolism, glucose homeostasis, and trophoblast differentiation. EDCs, such as phthalates, can act as agonists or antagonists of PPARs, disrupting their activity and subsequently altering miRNA expression. This disruption exacerbates the imbalance in angiogenic factors, oxidative stress, and endothelial dysfunction observed in PE [14]. Furthermore, these effects are not limited to the placenta; miRNA profiles in maternal blood and umbilical cord blood indicate systemic impacts of EDC exposure that may have long-term health implications for both mother and child.

The interplay between EDCs, miRNAs, and epigenetic modifications underscores a complex network of interactions that drive the progression and pathophysiology of PE. EDCs can initiate a cascade of events beginning with receptor-mediated signaling, leading to downstream epigenetic changes such as histone acetylation or methylation. These modifications, in turn, alter the expression of miRNAs and their target genes, ultimately impacting pathways critical for placental function and maternal–fetal health.

In addition to miRNA expression, DNA methylation changes observed in key placental genes, such as those regulating angiogenesis and immune responses, have been associated with EDC exposure. For example, hypermethylation of the VEGF promoter region reduces the expression of this critical pro-angiogenic factor, contributing to the anti-angiogenic state characteristic of PE.

The cumulative effects of these disruptions are profound. Placental insufficiency, oxidative stress, and systemic inflammation are all major hallmarks of PE that can be traced back to EDC-induced epigenetic alterations. This growing body of evidence highlights the need for a deeper understanding of how environmental exposure impacts the epigenome and initiates complications in pregnancy. Moreover, it underscores the potential of miRNAs and epigenetic biomarkers as tools for early diagnosis and targeted therapies for PE.

Many published reviews have established epigenetic mechanisms in the etiology of PE. However, there is significantly less evidence reporting the effect of EDCs on epigenetic alterations contributing to the development of PE. No systematic review has been conducted between EDCs and epigenetic alterations in the pathophysiology of PE. Therefore, by focusing on the intersection of endocrine disruption and epigenetics, we seek to highlight critical insights into the pathophysiology of PE, and the role of environmental exposure to these chemicals.

## 2. Methods

The guidelines followed were as per the Preferred Reporting Items for Systematic Reviews and Meta-Analyses (PRISMA) standards, and the systematic review methodology was registered with the International Prospective Register of Systematic Reviews (PROSPERO Registration number CRD42022338546). The checklist for PRISMA reporting is provided in a Appendix A PRISMA checklist [15].

### 2.1. Selection Criteria

We mainly incorporated epidemiological studies exploring exposure to EDCs (PBDEs, phthalates, BPA, PFNA, PFOA, PFOS, PFHxS) and epigenetic studies between PE and control pregnant women. Case-control, cross-sectional, prospective, and retrospective cohort studies meet the following inclusion criteria including the Peer-reviewed journal, research article, the English language, pregnant women, aged 18 to 45 years, gestational hypertension, PE, pregnancy-induced hypertension, endocrine-disrupting chemicals, endocrine disruptors, preterm birth, and fetal growth restriction, miRNA, and DNA methylation; (i) sample type-urine, blood, amniotic fluid, cord blood, or placenta; (ii) nested case-ontrol, cohort, retrospect, MIREC, or prospective and cross-sectional studies; (iii) epigenetic studies-small non-coding RNAs such as microRNAs and DNA methylation; (iv) signaling pathway-*PI3K/Akt/eNOS*, *MAPK/ERK*, and *VEGF* pathways.

The following are the exclusion criteria: non-peer-reviewed journal, review article, case reports and editorials, language other than English language, study not on humans, in vivo study, age < 14 or >45 years.

### 2.2. Search Strategy Description

A comprehensive search was performed on PubMed, PubMed Central, ScienceDirect, SCOPUS, ProQuest, Comparative Toxicogenomics Database (CTD), EBSCO host databases, and Google Scholar for studies between January 2010 and 30 July 2022. The search was performed with Boolean operations and the search terms used included the following: major search terms like “Preeclampsia”, “Endocrine disrupting chemicals”, and “microRNAs”, and a further deep search was performed, which included “ Preeclampsia”, “Pre-eclampsia”, “PE”, “eclampsia”, “pregnancy-induced hypertension or PIH”, “gestational hypertension”, “toxemia of pregnancy”, “edema proteinuria”, “hypertension ”, “Endocrine disrupting chemicals”, “EDC”, “EDCs”, “endocrine disruptors”, “EDs” “Phthalates”, “polybrominated diphenyl ether”, “PBDE”, “bisphenol A”, “BPA”, “triclosan”, “TCS”, “per fluorinated compounds”, “perfluorooctanoate”, “PFOS”, “per fluorinated octanoic acid”, “PFOA”, “perfluoroalkyl”, “PFA”, “perfluoro nonanoic acid”, “PFNA”, “xenoestrogen”, “DNA methylation”, “microRNA”, “miRNA”, “miRs”, “epigenetics”, and their synonyms. Additional filters added to the PubMed database search included: Human (women), full free text, adult: 18–44 years, English language, and last 10 years. 

Using the exclusion criteria (non-English, age > 45, animal models, unstructured abstract, published before 2000, full free text), 6859 papers were excluded. A total of 101 records were screened, and duplicate records were excluded (*n* = 52) using the Mendeley reference version 1.19.8 software. A total of 43 free full-text articles were assessed for eligibility, from which 14 articles were excluded as they had no exposure to selected EDCs or no health outcome measured (*n* = 8), so finally 29 studies were included in the qualitative data table synthesis.

### 2.3. Data Collection

All data were extracted and tabulated using a standardized data extraction method. Two authors independently reviewed the data, and any discrepancies were resolved in the presence of the corresponding author by consulting the relevant research articles. The preliminary results of the search were compiled. Further, 6 studies were found by manually searching the Google search engine and evaluating reference lists. All the records were screened based on the inclusion and exclusion criteria. By screening the titles and abstracts, additional relevant articles were added. The duplicate studies were removed with the assistance of the Mendeley reference manager. Free full-text articles from the search list meeting the inclusion criteria were included. Other inclusion criteria used were publication details (author, title, published year, and study location), methodological characteristics (study design, different matrix used, age, and sample size), exposure characteristics (type of EDCs, analysis method, and assays), and outcome assessment (95% confidence intervals or effect size with standard errors). The risk of bias (ROB) of the selected studies was evaluated by focusing on selection (application of inclusion and exclusion criteria and duration of the study), comparability (sample size similarity and age matching), and outcomes of the study (level of adjustment, exposure characterization, outcome assessment). The PECOS statement, including participants, exposure, comparators, outcomes, and study design, used to rate the studies for eligibility is reported in Table 1.

### 2.4. Data Extraction

The data were gathered from the included studies by using the following criteria: (i) age group; (ii) type of study; (iii) study population details; (iv) methodology used; and (v) type of sample. 

A total of 29 studies were qualified for qualitative synthesis, of which 11 were related to PE and EDCs, 10 were related to PE and miRNA, 8 were related to PE and DNA methylation, and finally, 2 studies described how prenatal exposure to EDCs alters miRNA expression in PE.

The studies were classified as case-control studies, cohort studies, or systematic reviews based on standardized tools established by the US National Toxicology Programs (NTPs), including the Office of Health Assessment and Translation (OHAT), the Newcastle–Ottawa Scale (NOS), the Integrated Risk Information System (IRIS), and the Toxic Substances Control Act (TSCA) of the United States Environmental Protection Agency (US-EPA). The ROB for all the included studies was checked by the two independent authors, responding to the following set of questions addressing the bias domains, with each question having three possible responses: “yes”, denoted as “+”; “no” denoted as “−”; and “not reported (NR)” (Table 1). Eight items were included in the ROB in the three perspectives (two items in selection, two items in comparability, and four items in outcome/exposure). 

## 3. Results

### 3.1. Study Selection and Characteristics

The numbers of articles selected for this systematic review from the different databases used are listed in Table 2. Quality assessments were analyzed and are presented in Table 3 (risk of bias); judgments of the author about the risk of bias in all included studies are presented. The total number of articles selected for the qualitative synthesis table in this review based on the eligibility criteria was 29. Out of the 29 articles, 11 articles reporting levels of EDC exposure are listed in Table 4. A summary of the literature analysis explaining the impact of EDCs and epigenetic alterations in PE is discussed in the data analysis table (Table 5).

The study selection is listed in the PRISMA flowchart (Figure 1). In these articles, the sample size characteristics range from 5 to 11,737. The majority of the studies reviewed comprised cohort studies (*n* = 13), case-control studies (*n* = 11), cross-sectional studies (*n* = 1), birth cohort studies (*n*= 2), and one study from the Agricultural Health Study (*n* = 1). Of the 29 studies identified, 12 were from the USA, 5 were from China, 2 each were from Sweden, Netherlands, and Norway, and there was 1 from each of Canada, Mexico, Taiwan, Czech Republic, Russia, and Turkey.

### 3.2. PE and Its Association with EDCs

Of the 12 studies, 11 showed an association between EDCs such as PFOAs, PBDE, phthalates, and phenolic compounds including BPA and triclosan (TCS) with PE and its related pregnancy disorders.

#### 3.2.1. Perfluoroalkyl Compound Exposure

Perfluoroalkyl substances (PFASs) are a class of synthetic chemicals such as PFOA, PFOS, PFBS, PFHxS, and PFUA used to produce non-stick vessels and water-resistant fabrics. A population-based study known as the C8 Health Studies (C8 refers to the eight-carbon chain of perfluorooctanoic acid) examined PFOA levels in a population exposed to chemical plants in Ohio and West Virginia since the 1950s. The exposure levels identified in this study were estimated to lead to a higher odds risk (OR) of stillbirth in the fourth quartile and the level of PFOA in serum was 4 ng/mL. The studies also showed a possible association of PFOA (OR = 1.00–1.28, 95% CI) exposure with PE compared to other adverse pregnancy outcomes in a highly exposed community [28]. 

A Norwegian cohort study conducted to evaluate the association of the levels of nine PFASs among nulliparous women showed no strong positive association between seven of the PFASs and PE, but a negative association was reported between PE and PFUA’s maximum quartile concentration relative to the minimum quartile (hazard ratio = 0.55, 95% CI = 0.38, 0.81) [30]. Another study conducted by this research group reported a positive association of PFASs with total cholesterol, which increased by 4.2 mg/dL per interquartile (IQ) shift. Five out of seven PFASs were positively associated with HDL cholesterol and all seven showed elevated HDL with the highest quartile of exposure. PFUnDA also showed the strongest positive association with HDL: HDL was increased by 3.7 mg/dL per IQ shift [29]. A cross-sectional study conducted in China reported positive associations of PFBS, PFHxS, and PFUA with the risk of PE and HDP, according to the elastic net penalty regression model, and a higher ln standardization of PFBS was associated with a higher AOR [2]. PFOS and PFNAs doubled with each IQ shift, demonstrating a correlation with an increase of PE by 38–53%. When confounding factors were adjusted, rising PFOS and PFNA exposure levels in serum were associated with a significant risk of early-onset PE (EOPE) [37]. Overall, these investigations demonstrated a strong association between prenatal PFAS exposure and PE risk and HDP.

Prenatal exposure to PFOA, PFOS, PFDA, PFNA, and PFUnDA is linked to low birth weight and small-for-gestational-age (SGA) births, with this association being stronger and statistically important among female neonates. These connections highlight concerns about societal well-being; however, further research is needed to determine the influence of sex and whether the effects persist throughout life. Because of the widespread exposure to chemicals and their extended half-lives in humans, preventive methods rather than phasing out may be impractical for PFASs with established detrimental health consequences [37].

A case-control study conducted among US populations showed that there are few associations of PFAS concentrations with angiogenic biomarkers (sFLt-1/PlGF), which are important hallmarks of PE [4].

#### 3.2.2. Phthalate Exposure

Phthalates are widely recognized to be the most common EDCs and are widely distributed in products used for preparing personal care products (PCPs), food processing products, plastics products, and adhesives. General exposure to these substances occurs in living beings through skin absorption, inhalation, or ingestion. Many studies have revealed that phthalate exposures might be linked with increased blood pressure (BP) and a higher risk for CVD [40]. In the NHANES survey, cross-sectional studies of the serum concentrations of monobenzyl phthalate (MBzP) metabolites were linked to several cardiovascular risk factors [26,34]. Researchers found a significant correlation between diastolic BP and the levels of mono-benzyl phthalate (MBzP) in the urine samples of pregnant women during the 20th week of gestation. Women in the third tercile for MBzP had diastolic blood pressure that was 0.5 mm higher (2.2 mm Hg higher) from the 16th week of gestation compared to the 20th week of gestation in the first tercile for MBzP. This study indicates that women had a higher chance of developing pregnancy-induced hypertensive (PIH) disorders including PE [34]. A study conducted in the Netherlands reported an association of bisphenols and phthalate metabolites with placental angiogenic markers and placental hemodynamics markers such as sFlt-1/PlGF [26].

#### 3.2.3. Phenolic Compound Exposure

Phenolic compound exposure includes exposure to BPA, which is commonly used in production of plastic water bottles, water supply pipes, children’s toys, thermal receipts, epoxy gums, and polycarbonate polymers. Triclosan (TCS), an antibacterial ingredient, is another phenolic EDC present in soaps and sunscreen lotions [41]. At the 10th week of gestation, an increase in IQ was associated with an increase in BPA (1.53; 95% CI: 1.04–2.25) and MEP (1.72; 95% CI: 1.28–2.3) concentration, contributing to the development of PE. Additionally, throughout the gestation period, all DEHP metabolites showed significantly elevated hazard ratios. This suggests that the urinary metabolite concentration of these phenolic compounds might be significantly linked with a high risk of PE, which in turn varies depending on fetal gender [20].

Increased clinical correlations between sFlt-1/PlGF and phthalate exposure, particularly between BPA and placental hemodynamics, may increase the risk of poor pregnancy outcomes. Hence, these studies showed an association between prenatal EDC exposure and increased risk of PE as compared to GHD [26].

After entering the bloodstream, several environmental factors reach the placenta and accumulate in trophoblast cells. This leads to mitochondrial damage in trophoblasts, impairing their vascular remodeling capacity. Consequently, inflammatory factors and other biological mediators are released, triggering significant placental inflammation, heightened systemic inflammation, oxidative stress, and an intensified immune response. These processes contribute to widespread organ damage, including severe placental dysfunction. The dysfunction is marked by disrupted trans-placental oxygen exchange, reduced blood flow, and an imbalance in angiogenic and anti-angiogenic factors, such as VEGF, PlGF, sFlt-1, and sENG, ultimately culminating in the development of PE [42].

### 3.3. Epigenetic Process Linked to PE

The epigenetic process might be influenced by various environmental stressors such as chemical exposure, PCPs, developmental abnormalities, dietary habits, etc. There are three classes of the epigenetic process: DNA methylation, non-coding miRNA expression, and histone methylation [9].

#### 3.3.1. DNA Methylation

Aberrant DNA methylation during placental development is one of the essential epigenetic processes correlated with PE. Studies have shown that dysregulation in methylation occurs both in early- (EOPE) and late-onset PE (LOPE). The first genomic mapping of methylation and hydroxymethylation of DNA in the human placental sample of LOPE suggests that the gene PTPRN2 and the ErbB signaling pathway may be important in the pathophysiology of LOPE [39]. Genes such as GRIN2b, GABRA1, PCDHB7, and BEX1 were differentially methylated and significantly associated with seizures in PE. Dysregulated membrane genes are associated with preterm delivery (PTM), premature rupture of membranes, fetal growth restriction, and neurodevelopmental disorders. PE placentas and early-trimester trophoblasts are significantly associated with altered gene expression. This reveals that epigenetic alterations in early pregnancy might cause defects in trophoblast function, contributing to various complications in PE. Moreover, the alterations in leukocyte DNA methylation in the maternal circulation are associated with PE during delivery and specific neural genes involved in preeclampsia risk [35].

POMC, AGT, CALCA, and DDAH1 are the candidate genes associated with CpG differential methylation, where DNA from leukocytes was observed in women with PE during delivery. Hence, differential methylation discovered in these candidate genes may play a role in the changes in gene expression and protein levels, as previously observed in PE [36]. In placental tissue, 5hmC and 5mC showed genome-wide distributions among LOPE patients and normal pregnancies. Significant alterations in 5hmC and 5mC may serve different roles in LOPE, each potentially contributing uniquely to its progression. The discovery of PTPRN2 and the enrichment of the ErbB signaling pathway exposed a relevant epigenetic mark for the LOPE etiology [39]. 

#### 3.3.2. PE and miRNA Alteration

The altered expression of non-coding miRNA expression is identified as one of the other epigenetic processes in PE, evidenced by many studies. Notably, the C19MC and C14MC clusters show elevated expression in the maternal circulation of the preeclamptic placentas. Since these clusters are placental-specific miRNAs, they are highly expressed from the early trimester to the late trimester. The placental samples of women with PE showed increased expression of miR-517a/b and miR-517c, belonging to the C19MC clusters. The heightened expression of these miRNA isoforms is associated with reduced trophoblast invasion and increased TNFSF15 expression and sFLT1 release [17]. miR-210-5p, a hypoxia miR, and its putative targets APLN and C3AR1 showed potential impact on the placenta in EOPE causing IUGR [18]. Genomic and epigenomic alterations studied in both placenta and trophoblast mainly focused on the TGF-ß pathway and showed common expression of the miRNAs miR-26a and miR-155 [19]. According to a cohort study conducted in Taiwan, miR-346 and miR-582-3p were associated with PE, PTM, and SGA [10]. Another cohort study from Russia reported downregulation of miR-135b and miR-98 and upregulation of miR-4532 in placental samples [31]. Certain miRNAs from the C14MC and C19MC clusters have been associated with pregnancy outcomes and complications including PTB and PE [11]. The altered microRNA (miRNA) expression pattern in the placenta was linked to PE, indicating that miRNA dysregulation contributes to the pathophysiology of PE.

#### 3.3.3. Effect of EDCs and Its miRNA Alteration in PE

Alteration in miRNA expression within the placenta predicted the association of prenatal exposure to phenols and phthalates, revealing the levels of EDC toxicity in humans. Concentrations of phenols and phthalates in urine samples positively associated with miRNA alteration in the placenta, including miR142-3p, miR-15a-5p, and miR-185 in the early trimester. In silico predictions by the researchers identified potential mRNA targets of these microRNAs, revealing their association with various biological processes. A Gene Ontology (GO) enrichment analysis indicated their involvement in regulating protein serine/threonine kinase activity [23]. Bicarbonate transport, hemoglobin metabolism, iron homeostasis, and small-molecule metabolism are four GO biological processes that are significantly correlated with the expression of these miRNAs related to EDC load. This is unique in explaining the involvement of EDCs and their miRNA alterations in the pathophysiology of PE [23]. Another study found an association between miRNAs such as miR-9-5p, miR-16-5p, miR-29a-3p, and miR-330-3p in sera of women with gestational diabetes mellitus (GDM), and PE with urinary levels of phthalate and phenol metabolites such as MEHP, MiBP, MBzP, MBP, and BPA. Phthalate analytes were detected in 97–100% of patients from urine samples, while the presence of BPA was found in only 40% of samples [24].

miRNAs are very constant and differentially expressed in pregnancy samples. Disrupted miR-346 and miR-582-3p expression profiles in maternal samples of plasma, fetal cord plasma, and placenta have been linked to PE and other pregnancy problems [10]. In a comparison of placental tissues from preterm premature rupture of membrane (PPROM) and gestational age-matched PTB pregnancies, several C19MC microRNAs including miR-516b-5p, miR-517-5p, miR-518f-5p, miR-518b, miR-519a, miR-520a-5p, miR-519d, miR-519e-5p, miR-520h, miR-525-5p, and miR-526, were found to be downregulated around the central cotyledon of the placenta in patients with PE [11]. Additionally, miR-524-5p showed reduced expression in PPROM pregnancies.

Intracellular kinases, including those in the MAPK and ERK signaling, play a significant role in PE by regulating cell proliferation, survival, and metabolic processes. This signaling may assist in metabolic functions impaired by PE’s altered metabolic syndrome. The MAPK/ERK signaling pathway was found to play a role in human villous trophoblast differentiation and invasion in the placenta, both of which were related to PE. miR-141-5p affects the MAPK1/ERK2 signaling pathway by controlling ATF2 via DUSP1, promoting the PE condition [33].

#### 3.3.4. Effect of EDCs and Genomic Imprinting

Researchers analyzed urine samples from pregnant women during the first trimester to assess the presence of eight phenol and eleven phthalate metabolites. Pyrosequencing for methylation of differentially methylated areas found significant associations between genes including H19, IGF2DMR0, and IGF2DMR2 and phenol and phthalate metabolites. Prenatal exposure to phthalate and phenol may disrupt the placental methylation of the H19 and IGF2 imprinting genes. Thus, alterations in DNA methylation from the exposure may be sexually dimorphic and EDC-specific. Specifically, prenatal exposure to phthalate metabolites such as HMW and DEHP was linked to abnormal H19 imprinting in male neonates [22]. Alterations in placental DNA methylation are part of the biological triggering mechanism linking prenatal PBDE exposure to poor fetal development. HSD11B2/IGF2 methylation was found to be associated with the development of fetal growth retardation (FGR). PBDE congeners such as BDE-17-190 showed significant positive results in placental methylation, and 19 BDE congeners were measured in the cord blood samples. The combined concentration of the 19 PBDEs was 51.354 ng/g lw, exceeding the levels documented in earlier studies [38].

Another study found that high total phthalate exposure causes placental EGFR (18/51 pathways), hypermethylation, and reduced expression in pregnant women, indicating that it is one of the key pathways and a potential therapeutic target for phthalate disruption in the endocrine system. To our knowledge, this is one of the first studies to investigate the impact of phthalates on the expression and methylation of human-imprinted placental genes, and it focuses on the first trimester of pregnancy. In the high-phthalate-exposure group, where the average concentration was 231 ng/mL, substantial changes in DNA methylation were observed and 39 genes were expressed, of which the most notable one was EGFR [3].

## 4. Discussion

Based on the findings of this systematic examination, most of the epidemiological studies included showed that women exposed to EDCs such as phenolic compounds, phthalates, PBDE, and PFAS are at higher risk of reproductive abnormalities. The main sources of exposure were open-burning dump yards, usage of day-to-day personal care products, polluted water from industries, and packaged foods. Exposure to these EDCs leads to pregnancy-related outcomes such as stillbirth, SGA, FGR, EOPE, increased cholesterol, obesity, CVD, and PIH. Several studies included in the systematic review supported an association between PE and differential miRNA expression. In addition to circulatory miRNAs, dysregulation of placental miRNAs including the C19MC and C14MC miRNA clusters has also been observed in PE. The major pathways regulated by these miRNAs were angiogenesis, EGFR, TGF-ß, and MAPK/ERK, which are linked to cell proliferation, survival, metabolism, trophoblast differentiation, and invasion in the placenta. Altered genomic methylation was also reported in the placental samples of EOPE and LOPE, leading to PTM, mainly due to defects in trophoblast function. Out of the selected 29 articles, only two articles reported an association of EDC exposure with altered miRNA expression leading to the pathophysiology of PE. However, in-depth studies are required to elucidate the mechanism behind the altered miRNA expression caused by EDCs in PE, its subtypes, and other developmental abnormalities in the fetus.

The review also highlights a significant association between angiogenic markers and the effects of EDCs in PE. Limited but noteworthy evidence suggests that EDCs such as BPA, phthalates, and PFOA disrupt pro-angiogenic signaling, contributing to an imbalance between sFlt-1 and PlGF. Additionally, BPA and hypoxia have been shown to upregulate miR-210, which inhibits mitochondrial function and angiogenesis by downregulating VEGF. Similarly, phthalates and dioxins suppress miR-126 expression, impairing endothelial cell survival and angiogenic processes. These miRNA alterations potentiate anti-angiogenic signaling, oxidative stress, and inflammation, thereby exacerbating placental dysfunction.

Furthermore, phthalates induce hypomethylation of pro-inflammatory genes such as TNF-α and IL-6, amplifying systemic inflammatory responses. Collectively, these epigenetic modifications, encompassing DNA methylation and miRNA dysregulation, contribute to impaired placental function, increased oxidative stress, and endothelial dysfunction, which are hallmark features of PE. Future studies should focus on the epigenetic mechanisms underlying placental dysfunction in PE, as this may provide critical insights into the pathophysiology and potential therapeutic targets for the disorder.

This systematic review will further help to ascertain feasible biomarkers for the early identification of the disease and facilitate the identification of therapeutic targets for the formulation of treatment techniques for PE caused by exposure to EDCs. According to multiple pieces of evidence, early-life exposure to environmental pollutants, even at exceptionally low concentrations, has long-lasting effects and leaves epigenetic marks over a generation. Moreover, these phthalate and phenolic exposures might also be associated with gestational hypertensive diseases such as PE, HELLP, etc. These findings will help investigators find preventive measures to minimize the population’s exposure to various forms of EDCs during the prenatal period.

## Figures and Tables

**Figure 1 cells-14-00493-f001:**
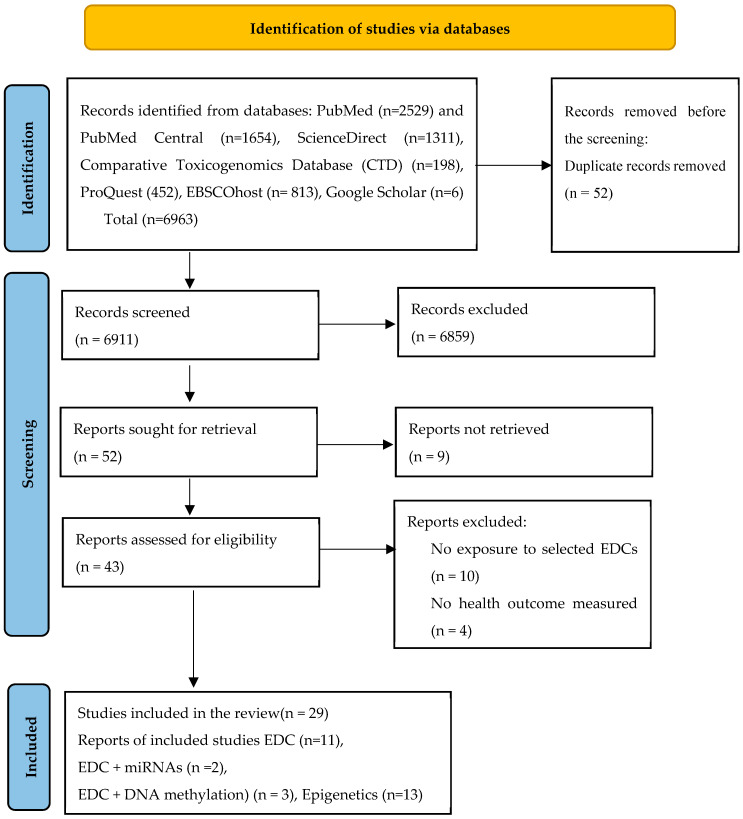
Flow diagram showing the process of study selection for inclusion in the systematic review.

**Table 1 cells-14-00493-t001:** Selection criteria according to PECO statement.

Category	Criteria
Population	Preeclampsia patient group
Exposure	One of the following endocrine disrupters (“Phthalates”, “PBDE”, “bisphenol A”, “triclosan”, “TCS”, “PFOS”, “PFOA”, “PFNA”)
Comparison	Healthy mothers with patient groups
Outcome	Epigenetic alterations including miRNAs, DNA methylation
Study design	Observational studies (cohort, case–control, cross-sectional)

**Table 2 cells-14-00493-t002:** Systematic reference strategies used in different databases.

	PubMed	PubMed Central	ScienceDirect	ProQuest	CTD	EBSCOhost	Google Scholar (Hand Search)
Preeclampsia OR Pre-Eclampsia OR eclampsia OR PE OR pregnancy-induced hypertension OR PIH OR gestational hypertension OR toxaemia of pregnancy OR edema proteinuria OR hypertension gestos	99,469	462,776	123,843	48,871	103	164,222	
Endocrine disrupting chemicals OR EDC OR EDCs OR endocrine disruptors OR EDs OR Phthalates OR polybrominated diphenyl ether OR PBDE OR bisphenol A OR BPA OR triclosan OR TCS OR per fluorinated compounds OR perfluorooctanoate OR PFOS OR per fluorinated octanoic acid OR PFOA OR perfluoroalkyl OR PFA OR perfluoro nonanoic acid OR PFNA OR xenoestrogen	100,313	421,370	119,334	47,755	895	14,482	
DNA methylation OR microRNAs OR miRNA OR miRs OR epigenetics	315,176	513,992	152,976	26,128	985	1329	
Preeclampsia AND endocrine disrupting chemicals AND microRNAs	2	145	58	2	1	1	6
Further screening	103						
After duplicates removed	51						
Qualitative synthesis	29						

**Table 3 cells-14-00493-t003:** Risk of bias: Judgments of the author about the risk of bias in all included studies represented by “+”, “−”, and “NR”.

		Selection		Comparability		Outcome			
No.	References	Were the Inclusion and Exclusion Criteria Applied to All Groups?	Were All the Studies Performed over the Same Time Period?	Were the Sample Size of All Groups Similar?	Were the Groups Age-Matched?	Level of Adjustment (Analysis/Design)	Were All the Studies Confident in the Exposure Characterization?	Were All the Studies Confident in the Outcome Assessment?	Did Authors Report Conflicts of Interest?
1	[16]	+	+	−	−	+	−	+	+
2	[17]	+	+	−	−	+	−	+	+
3	[18]	+	−	−	−	+	−	−	+
4	[4]	+	−	−	−	+	+	+	+
5	[19]	NR	NR	+	−	+	+	+	+
6	[20]	+	−	−	−	+	+	+	+
7	[3]	+	−	−	−	+	+	+	+
8	[21]	NR	NR	+	−	NR	NR	+	NR
9	[11]	+	−	−	−	+	−	+	+
10	[2]	+	−	−	−	+	+	+	+
11	[22]	NR	−	−	−	+	+	+	NR
12	[23]	NR	−	−	−	+	+	+	−
13	[24]	+	−	−	−	+	+	+	+
14	[25]	+	+	−	−	+	+	+	+
15	[26]	+	−	−	−	+	+	+	+
16	[27]	+	−	−	−	−	+	+	NR
17	[28]	−	−	−	−	+	+	+	−
18	[29]	+	−	−	−	+	+	+	+
19	[30]	+	−	−	−	+	+	+	−
20	[10]	+	−	−	−	+	−	+	+
21	[31]	+	NR	−	−	+	−	+	NR
22	[32]	+	−	−	−	+	−	+	+
23	[33]	+	−	+	−	+	−	+	+
24	[34]	+	−	−	−	+	+	+	+
25	[35]	+	−	+	−	+	−	+	+
26	[36]	+	−	+	−	+	−	+	+
27	[37]	+	−	−	−	+	+	+	+
28	[38]	+	−	−	−	+	+	+	+
29	[39]	+	−	+	−	+	−	+	+

**Table 4 cells-14-00493-t004:** EDC exposure level and the outcomes in preeclampsia.

No.	Author	Sample Size	Sample/Methodology	Location	Exposure	Outcome
1	[3]	49/34	Placenta/SPE-HPLC isotope dilution–tandem mass spectroscopy	USA	TP (total phthalate) concentration = 231 ng/mL.	Phthalates impact placental function by modulating the expression of critical placental genes through epigenetic regulation.
2	[2]	42/644	Plasma/LC system coupled with tandem mass spectrometry	China	Plasma concentrations of PFOA = 6.98 ng/mL, PFOS = 2.38 ng/mL, PFNA = 0.64 ng/mL, PFUA = 0.40 ng/mL, PFDA = 0.36 ng/mL, PFHxS = 0.16 ng/mL, PFDoA = 0.094 ng/mL, and PFBS = 0.047 ng/mL.	Investigates the association between prenatal PFAS exposure and hypertensive disorders of pregnancy (HDP) in humans.
3	[23]	179	Placenta/SPE-HPLC isotope dilution–tandem mass spectroscopy	United States	Phthalate metabolite concentrations: MECPP = 0.6 μg/L, MEHHP = 0.7 μg/L, MEOHP = 0.7 μg/L, MEHP = 1.2 μg/L, MCPP = 0.2 μg/L, MCOP = 0.7 μg/L, MCNP = 0.6 μg/L, MBzP = 0.3 μg/L, MiBP = 0.6 μg/L, MnBP = 0.6 μg/L, and MEP = 0.8 μg/L. Phenol concentrations: DCP = 0.2 μg/L, 2,5-DCP = 0.2 μg/L, BP-3 = 0.4 μg/L, BPA = 0.4 μg/L, BuPB = 0.2 μg/L, MePB = 1.0 μg/L, PrPB = 0.2 μg/L, and TCS = 2.3 μg/L.	miR-142-3p, miR15a-5p, and miR-185 are the three miRNAs significantly linked to phenol or phthalate levels influencing their expression in the placenta.
4	[25]	73	Serum/SPE-HPLC isotope dilution–tandem mass spectroscopy	Sweden	Serum concentrations of PFNA = 0.7 ng/mL, PFOA = 1.8 ng/mL, PFOS = 5.6 ng/mL, and PFHxS = 1.9 ng/mL.	Serum concentrations of PFNA, PFOA, and PFOS show an inverse association with kidney function; change is unrelated to parallel changes in eGFR and glomerular pore size.
5	[26]	64/1155	Placenta/SPE-HPLC isotope dilution–tandem mass spectroscopy	The Netherlands	Phthalate metabolite concentration = 0.19 ng/mL.	No consistent associations of early-pregnancy bisphenol and phthalate metabolite concentrations with maternal prenatal BP, placental hemodynamic outcomes, or gestational hypertensive disorders.
6	[28]	11,737	Serum/liquid chromatography–tandem mass spectrometry	USA	Serum concentrations of PFOA = 4 ng/mL.	No associations between estimated serum PFOA levels and adverse pregnancy outcomes other than possibly preeclampsia.
7	[29]	891	Plasma/HPLC system coupled with tandem mass spectrometry	Norway	PFOS concentration associated with cholesterol = 4.2 mg/dL.	Elevated HDL (high-density lipoprotein) is not an adverse outcome per se; elevated total cholesterol associated with PFASs during pregnancy could be of concern if causal.
8	[34]	369	Urine/isotope dilution HPLC coupled with tandem mass spectrometry	USA	Urine phthalate metabolite concentrations at 16 weeks gestation: ΣDBP (31 vs. 27 μg/g Cr), ΣDEHP (99 vs. 89 μg/g Cr), MCPP (2.5 vs. 2.7 μg/g Cr), MBzP (9.2 vs. 11.2 μg/g Cr), and MEP (139 vs. 187 μg/g Cr).	Maternal urinary (MBzP) phthalate concentrations may be associated with increased diastolic blood pressure and risk of pregnancy-induced hypertensive diseases.
9	[37]	64/1709	Serum/liquid chromatography-tandem mass spectrometry	Sweden	Serum levels of PFOS = 5.4 ng/mL, PFOA = 1.6 ng/mL, PFNA = 0.5 ng/mL, PFDA = 0.26 ng/mL, PFUnDA = 0.21 ng/mL, PFHxS = 1.32 ng/mL, PFHpA = 0.018 ng/mL (*p* = 0.06 for PFOS, *p* = 0.07 for PFOA, *p* = 0.25 for PFNA, *p* = 0.33 for PFDA, *p* = 0.77 for PFUnDA, *p* = 0.62 for PFHxS, and *p* = 0.83 for PFHpA).	Increasing serum levels of PFOS and PFNA during early pregnancy were associated with a clinically relevant risk of preeclampsia, adjusting for established confounders.
10	[38]	373/125	Placenta/GC tandem MS	China	Total PBDE (polybrominated diphenyl ether) concentration = 51.354 ng/g lw.	Changes in placental DNA methylation might be part of the underlying biological pathway between prenatal PBDE exposure and adverse fetal growth.
11	[24]	18/22	Urine/placenta	Mexico; US	Detection levels are not significant compared to normal non-diabetic women. However, phthalates and BPA urinary levels showed positive correlations between adjusted urinary MBzP levels and miR-16-5p expression levels (*p* < 0.05), and adjusted MEHP concentrations and miR-29a-3p expression levels (*p* < 0.05).	Serum levels of miRNAs associated with GDM (miR-9-5p, miR-16-5p, miR-29a-3p, and miR-330-3p) and urinary levels of phthalate metabolites (mono-n-butyl phthalate (MBP), mono-isobutyl phthalate (MiBP), mono-benzyl phthalate (MBzP), and mono(2-ethyl hexyl) phthalate (MEHP)) and bisphenol A in GDM and PE patients.

**Table 5 cells-14-00493-t005:** Data analysis table for preeclampsia: Summary of literature analysis explaining the impact of EDCs and epigenetic alterations in preeclampsia.

No.	Age Group	Type of Study	Study Population	Sample Size (Case/Control)	Sample	Method	EDC	Epigenetic Mechanism	Pathway	Other Findings	References
**a. Studies Reported on EDCs**											
1	Mean age of 32.2	LifeCodes prospective birth cohort case–control study population	United States	75/75	Plasma	Online solid-phase extraction (SPE) coupled to high-performance liquid chromatography (HPLC) isotope dilution–tandem mass spectrometry (LC-MS-MS); angiogenic markers were analyzed.	PFOA and PFDA	-	-	PFOA and PFDA were found to be associated with late-onset preeclampsia and angiogenic biomarkers like SFlt-1 and PlGF.	[4]
2	20–45 years	Cohort study	Sweden	73	Blood	PFASs were analyzed using LC-MS.	PFNA, PFOA, and PFOS	-	-	Serum concentration of PFNA, PFOA, and PFOS declined during pregnancy.	[25]
3	28–35 years	Case–control	Sweden	64/1709	Blood	Serum was analyzed using LC-MS.	PFOA and PFNA	-	Toxicological pathways related to inflammation and oxidative stress.	PFOS and PFNA exposures were significantly associated with preeclampsia.	[37]
4	20–45 years	Cross-sectional study	China	42/644	Umbilical cord blood	LC-MS.	PFBS	-	-	PFBS disrupts the regulation of estrogen levels. PFBS exposure associated with preeclampsia.	[2]
5	20–44 years	Cohort study	Netherlands	64/1155	Urine	HPLC–T-MS analysis and sFlt-1 and PlGF concentrations were measured using an immune-electrochemoluminence assay.	Phthalates	-	-	Phthalates exposure elevate clinical associations with the sFlt-1/PlGF ratio.	[26]
6	27–37 years	A nested case–control study	United States	50/431	Urine	BPA and phthalate concentrations were measured based on methods developed by the Centers for Disease Control (CDC).	BPA and Phthalate	-	-	Urinary BPA and phthalate concentrations are associated with preeclampsia.	[20]
7	23–35 years	Cohort study	United States	369	Urine	The samples were analyzed by isotope dilution HPLC-T-MS.	MBzP	-	-	MBzP concentrations were associated with blood pressure-related complications.	[34]
8	19–44 years	Cohort study	Norway	891	Blood	PFASs were analyzed using HPLC-T-MS.	PFAS	-	-	PFASs were positively associated with elevated cholesterol level.	[29]
9	16–44 years	Cohort study	Norway	466/510	Blood	PFASs were measured using HPLC-T-MS.	PFAS	-	-	PFAS exposure was not strongly associated with preeclampsia.	[30]
10	14–45 years	Cohort study	United States	11,737	Blood	PFOAs were analyzed using LC-T-MS.	PFOA	-	-	No associations between estimated serum PFOA levels and adverse pregnancy outcomes other than possibly preeclampsia.	[28]
11	16–49	The Agricultural Health Study	United States	11,274	From self-reported data	Regression analysis was performed on agricultural and pesticide exposure activities among farmers involved in pesticide application.	Pesticides	-	-	First-trimester residential and agricultural activities with potential exposure to pesticides were associated with both PIH and PE.	[27]
**b. Studies Reported on Epigenetics**											
1	29–35 years	Cohort study	Canada	58/21	Placenta	RNA sequencing on the Illumina Hiseq2000 platform.	-	miR-210-5p	-	miR-210-5p was downregulated and the gene targets such as APLN and C3AR1 were downregulated in placental samples.	[18]
2	28–31 years	Cohort study	China	30/30	Placenta	“TargetScan Human Release 7.2”.	-	miR-141-5p	p-MAPK1 and ERK1/2 signaling.	miR-141-5p has the potential to regulate ATF2, promoting the expression of phosphatase DUSP1, which in turn affects p-MAPK1 and ERK1/2 signaling, contrubuting to the development of preeclampsia.	[33]
3		A nested case–control study	Czech Republic	PE-21, IUGR-18/control-58	Plasma	miRNA analysis using real-time PCR.	-	miR-516b-5p, miR-517-5p, miR-520a-5p, miR-525-5p, and miR-526a	Eph/ephrin signaling, calcium influx, phosphoinositide 3 (PI3) kinase, mitogen-activated protein (MAP) kinase, Src kinase, and Rho.	Abnormal levels of extracellular fetal DNA, mRNA transcripts, and circulating C19MC microRNAs (miR-516b-5p, miR-517-5p, miR-520a-5p, miR-525-5p, and miR-526a) have been associated with pregnancy complications.	[11]
4	29–35 years	Cohort study	Taiwan	79/60	Blood and placenta	qRT-PCR.	-	miR-346 and miR-582-3p	-	miR-346 and miR-582-3p were associated with preeclampsia, preterm delivery (PTM), and SGA.	[10]
5	-	Case–control	United States	16/16	Placenta	Placental mi-RNAs were analyzed by qRT-PCR.	-	miR-26a and miR-155	TGF-ß pathway.	Genes within the TGF-ß pathway displayed increases in PE placenta and Cd-treated trophoblast, and its targets miR-26a and miR-155 were found to be significantly altered.	[19]
6	29–37 years	Cohort study	Russia	05/06	Placenta	qPCR.	-	miR-135b, miR-98, and miR-4532	Hypoxia pathways.	miR-135b and miR-98 were downregulated and miR-4532 was upregulated in placental samples.	[31]
7	19–34 years	A case–control study	China	31/14	Placenta	miRNA expression using qPCR.	-	miR-517a/b and miR-517c	Tumor necrosis factor super family 15 (TNFSF15) signaling pathway, NF-kB and MAPK signaling pathways.	miR-517a/b and miR-517c were dysregulated in the placenta, which increased production of anti-angiogenic cytokines such as TNFSF15 and sFLT1. Several genes of miR-517a/b/c, including HOXA5, SEMA3A, TFAP2B, and PTK2B, play a role in cell invasions.	[17]
8	-	Case–control study	Turkey	20/20	Blood/	miRNA expression using qPCR.	-	miR-210 and miR-152	-	miR-210 and miR-152 were found to be altered in PE.	[21]
9	19–27 years	Cohort study	United States	124/125	Blood	Illumina Human Methylation-27 Assay.	-	DNA methylation	-	Differential methylation of POMC, AGT, CALCA, and DDAH1 genes; CpG sites associated with preeclampsia.	[36]
10	23–32 years	Case–control	Netherlands	76/76	Placenta	MassARRAY EpiTYPER assays.	-	DNA methylation	ErbB signaling pathway, p53-pathway, type-Ⅰ diabetes mellitus pathway.	5hmC and 5mC changes may play a vital role in the development of late-onset preeclampsia.	[39]
11	20–35 years	A prospective case–control study	United States	31/14	Placenta	DNA methylation using genome-wide Illumina Infinium Methylation 450 Bead Chip array.	-	DNA methylation	Placental gene pathways	Changes in CDH11, COL5A1, NCAM1, and TNF gene expression were associated with alteration in methylation which contributed to changes in the placenta.	[16]
12	25–30 years	Case–control	United States	14/14	Blood	Illumina Human Methylation-27 Assay.	-	DNA methylation	Neuropeptide signaling pathway	Methylation changes were detected in the genes of maternal leukocyte DNA in preeclamptic pregnancies at the time of delivery.	[35]
13	28–41 years	Cohort study	China	30/32	Placenta	Illumina Infinium HumanMethylation450 (450k) Bead Chip.	-	DNA methylation	-	Hypomethylated in the placental DNA methylome may lead to impaired function of de novo DNA methyltransferases, disrupting trophoblast turnover and potentially contributing to the development of PE.	[32]
**c. Studies Reported on EDCs and Epigenetic Changes**											
1	22–32 years	Case–control	China	124/125	Umbilical cord	Serum was analyzed using liquid chromatography–tandem mass spectrometry.	PBDE	DNA methylation	-	HSD11B2/IGF2 methylation due to PBDE exposure was associated with the development of fetal growth retardation.	[38]
2	24–45 years	Case–control study	Mexico City, Mexico	18/22	Urine	Urinary phthalate metabolites were analyzed using UPLC-MS/MS. miRNA expression was analyzed by qRT-PCR.	phthalate metabolites MBP, MiBP, MEHP, MBzP, and BPA	miR-9-5p, miR-16-5p, miR-29a-3p, and miR-330-3p	MAPK, insulin, TGF-β, and mTOR signaling pathways, glycolytic pathways.	Serum levels of miRNAs (miR-9-5p, miR-16-5p, miR-29a-3p, and miR-330-3p) were associated with GDM and PE. Urinary levels of the phthalate metabolites MBP, MiBP, MEHP, MBzP, and BPA were analyzed in GDM and PE patients.	[24]
3	19–49 years	Cohort study	United States	49/34	Urine/placenta	Urinary phthalate metabolite concentrations using SPE-HPLC-T-MS. Placenta methylation using Illumina’s Infinium Human Methylation 850k BeadChip.	Phthalate	DNA methylation	ErB signaling pathway.	Phthalate exposure was associated with first-trimester DNA methylation changes associated with EGFR.	[3]
4	27–37 years	Cohort study	United States	179	Placenta	Placental miRNAs were analyzed by qRT-PCR.	Phthalate and phenol	miR-185, miR-142-3p, miR15a-5p	Protein serine/threonine kinase, positive regulation of protein insertion into mitochondrial membrane involved in apoptotic signaling pathway, IGFR signaling pathway, PlGFR signaling pathway.	Urinary phthalate and phenol exposure were associated with alteration of miRNA (miR-185, miR-142-3p, miR15a-5p) expression.	[23]

## Data Availability

No new data were created or analyzed in this study.

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
