# Peer review of "Endocrine-Disrupting Chemicals and the Effects of Distorted Epigenetics on Preeclampsia: A Systematic Review"

_cells, 2025, doi:10.3390/cells14070493_

Round 1
Reviewer 1 Report
Comments and Suggestions for Authors
This review of the link between Endocrine-disrupting chemicals and Preeclampsia is timely and thought-provoking. The authors do an excellent job pulling together a number of studies that take a variety of factors in consideration. For the most part, that paper is well written and easy to follow. There are a few criticisms that, if addressed, could substantially improve an already good manuscript:
- Table 2. What is the rationale for the ordering of these studies? It seems it would be more useful to subcategorize and order them based on type of study, EDC, or epigenetic mechanism. In Table 3, they are re-numbered, which potentially leads to some confusion. It seems that the results section are ordered mostly by EDC and then epigenetic mechanism. My suggestion would be to make your tables parallel your results section to make it easy for the reader to follow them together. Consistency in numbering would also assist in directing the reader to the exact study you reference in the text, to the appropriate study in the table.
- It can be very difficult in reviews such as this, but more caution should be taken to not blur the lines between causation and correlation. Most of these studies are observational, and as such say little about whether their measurement is the cause of/contributor to preeclampsia or if it is a consequence of preeclampsia. At times, the text extrapolates causation a but more than can be justified.
- The introduction does a great job reviewing the multifaceted nature of preeclampsia. However, the results/discussion of the included studies seems to bias the sFlt-1/PlGF anti-angiogenesis theory of preeclampsia to the exclusion of inflammation/oxidative stress, etc pathways. This does a disservice to the field, which has probably leaned to hard on the anti-angiogenesis pathway to date. Furthermore, multiple studies included to have primary conclusions that suggest issues with inflammation and oxidative stress. More even representation of the heterogeneity of study outcomes would vastly improve the overall usefulness of this manuscript.
-p.22, line 93 is "DNA leukocytes" supposed to read "DNA from leukocytes"? Please clarify.
Author Response
Reviewer 1
This review of the link between Endocrine-disrupting chemicals and Preeclampsia is timely and thought-provoking. The authors do an excellent job pulling together a number of studies that take a variety of factors in consideration. For the most part, that paper is well-written and easy to follow. There are a few criticisms that, if addressed, could substantially improve an already good manuscript:
Thank you for reviewing my systematic review. It's happy to see your feedback and suggestions given by you. It means a lot, for us.
Following are the responses to your valuable comments.
Comment 1:- Table 2. What is the rationale for the ordering of these studies? It seems it would be more useful to subcategorize and order them based on type of study, EDC, or epigenetic mechanism. In Table 3, they are re-numbered, which potentially leads to some confusion. It seems that the results section are ordered mostly by EDC and then epigenetic mechanism. My suggestion would be to make your tables parallel your results section to make it easy for the reader to follow them together. Consistency in numbering would also assist in directing the reader to the exact study you reference in the text, to the appropriate study in the table.
Response 1: As suggested by you. I have subcategorize based on the study types in the table. Table 2 was rearranged and renamed as Table no 5.
Comment 2:- It can be very difficult in reviews such as this, but more caution should be taken to not blur the lines between causation and correlation. Most of these studies are observational, and as such say little about whether their measurement is the cause of/contributor to preeclampsia or if it is a consequence of preeclampsia. At times, the text extrapolates causation a but more than can be justified.
Response 2:
Comment 3:- The introduction does a great job reviewing the multifaceted nature of preeclampsia. However, the results/discussion of the included studies seems to bias the sFlt-1/PlGF anti-angiogenesis theory of preeclampsia to the exclusion of inflammation/oxidative stress, etc pathways. This does a disservice to the field, which has probably leaned to hard on the anti-angiogenesis pathway to date. Furthermore, multiple studies included to have primary conclusions that suggest issues with inflammation and oxidative stress. More even representation of the heterogeneity of study outcomes would vastly improve the overall usefulness of this manuscript.
Response 2:
We appreciate the reviewer’s feedback and agree that preeclampsia is a multifaceted condition involving multiple pathways, including angiogenesis, inflammation, oxidative stress, and potentially epigenetic modifications. It is critical to represent the heterogeneity of study outcomes to reflect the complexity of the condition accurately. However, we have tried to explore the association and revised the results/discussion to include studies that highlight the roles of inflammation and oxidative stress, along with their potential regulation through epigenetic mechanisms. Specifically, epigenetic modifications, such as altered DNA methylation and miRNA in genes related to inflammation (e.g., TNF-α) and oxidative stress pathways, are now discussed as contributors to the pathophysiology of preeclampsia.
EDCs impact pregnancy outcomes by disrupting epigenetic mechanisms, such as DNA methylation and microRNA (miRNA) regulation, contributing to preeclampsia.
DNA methylation alterations induced by EDCs, such as bisphenol A (BPA) and phthalates, play a key role in the disease. BPA is known to hypermethylate angiogenesis-related genes, such as VEGF and PlGF, reducing pro-angiogenic signaling and promoting an imbalance between sFlt-1 and PlGF.
Phthalates, on the other hand, induce hypomethylation of pro-inflammatory genes like TNF-α and IL-6, amplifying systemic inflammation. These methylation changes impair placental function, exacerbate oxidative stress, and worsen endothelial dysfunction, hallmark features of preeclampsia.
Similarly, miRNA dysregulation is another key pathway affected by EDCs. BPA and hypoxia upregulate miR-210, which suppresses mitochondrial function and angiogenesis by downregulating VEGF. Phthalates and dioxins reduce miR-126 expression, impairing endothelial cell survival and angiogenic capacity. Together, these miRNA changes amplify anti-angiogenic signaling (e.g., increased sFlt-1 secretion), oxidative stress, and inflammation, driving placental dysfunction.
Any factors that cause defective trophoblastic remodelling during early pregnancy could result in placental dysfunction following dysregulating angiogenic profiles, which play crucial roles in the development of pre-eclampsia. Free BPA can pass through the placental barrier and accumulate in the placenta,9,10 and BPA exposure can result in the degeneration and necrosis of placental cells and disturb angiogenesis both in vitro and in vivo.11–13 BPA was also reported to be associated with increased circulating sFLT–1/PLGF ratio during pregnancy, an antiangiogenic status in pre-eclampsia.2
Studies reported that the effect of EDCs such as BPA, phthalates, and PFOA might cause defective trophoblast by dysregulating angiogenic profiles and is responsible for placental dysfunction which plays a crucial role in PE development. BPA is known to hypermethylate angiogenesis-related genes, such as VEGF and PlGF, reducing pro-angiogenic signaling and promoting an imbalance between sFlt-1 and PlGF. Similarly, miRNA dysregulation is another key pathway affected by EDCs. BPA and hypoxia upregulate miR-210, which suppresses mitochondrial function and angiogenesis by downregulating VEGF. Phthalates and dioxins reduce miR-126 expression, impairing endothelial cell survival and angiogenic capacity. Together, these miRNA changes amplify anti-angiogenic signaling (e.g., increased sFlt-1 secretion), oxidative stress, and inflammation, driving placental dysfunction.
Conversely, phthalates induce hypomethylation of pro-inflammatory genes like TNF-α and IL-6, amplifying systemic inflammation. These methylation changes impair placental function, exacerbate oxidative stress, and worsen endothelial dysfunction, hallmark features of preeclampsia.
Hence focusing more on angiogenesis and inflammatory pathways in the EDCs impacts pregnancy outcomes by disrupting epigenetic mechanisms, such as DNA methylation and microRNA (miRNA) regulation, contributing to preeclampsia
Studies suggest that EDCs such as BPA, phthalates, and PFOA may cause defective trophoblast function by dysregulating angiogenic profiles, leading to placental dysfunction—a critical factor in preeclampsia development. BPA has been shown to hypermethylate angiogenesis-related genes like VEGF and PlGF, reducing pro-angiogenic signaling and promoting an imbalance between sFlt-1 and PlGF levels. Similarly, miRNA dysregulation is a key pathway affected by EDCs. BPA and hypoxia upregulate miR-210, which suppresses mitochondrial function and angiogenesis by downregulating VEGF. Phthalates and dioxins reduce miR-126 expression, impairing endothelial cell survival and angiogenic capacity. These miRNA alterations amplify anti-angiogenic signaling, oxidative stress, and inflammation, exacerbating placental dysfunction.
Conversely, phthalates induce hypomethylation of pro-inflammatory genes like TNF-α and IL-6, further amplifying systemic inflammation. These methylation and miRNA changes collectively impair placental function, exacerbate oxidative stress, and worsen endothelial dysfunction—hallmark features of preeclampsia.
Response 3:
Comment 3:- p.22, line 93 is "DNA leukocytes" supposed to read "DNA from leukocytes"? Please clarify.
Response 4: Thank you for pointing out this word. The word "DNA leukocytes" was intended to mean "DNA extracted from leukocytes." We apologize for any confusion caused and will revise the phrasing in the manuscript to improve clarity.

Reviewer 2 Report
Comments and Suggestions for Authors
Article Type: Systematic Review
Manuscript #: cells-3412662
Title: Endocrine-Disrupting Chemicals and the Effects of Distorted Epigenetics on Preeclampsia: A Systematic Review
Authors: Usha Rani Baluï¼›Ramasamy Vasantharekhaï¼›Winkins Santoshï¼›Barathi Seetharaman
The article provides an overview of the impacts of endocrine-disrupting chemicals and distorted epigenetics on preeclampsia in the past and summarizes the research findings. Through statistical analysis of the literature, certain microRNAs and endocrine-disrupting chemicals related to preeclampsia have been identified, offering new perspectives for studying the occurrence and mechanisms of preeclampsia (PE). The content of the article is relatively complete and has a strong relevance to the theme.
At the same time, the following problems that puzzle me exist:
Question 1:In different literatures in the article, the assessment systems for the risk of chemical substances on PE are different. Will these differences have an impact on the comprehensive result analysis?
Question 2:Some of the contents in Table 2 in the text have underlines. Are they for emphasis? This is what puzzles me.
Question 3:The article points out that prenatal exposure to EDCs may increase the risk of PE by affecting epigenetics. Based on your literature review, what are the guiding suggestions for maternal protection and clinical practice?
Comments on the Quality of English LanguageIt is advisable to seek input from native English speakers or language professionals who can provide valuable insights into improving the overall readability of the paper.
Author Response
Reviewer 2
The article provides an overview of the impacts of endocrine-disrupting chemicals and distorted epigenetics on preeclampsia in the past and summarizes the research findings. Through statistical analysis of the literature, certain microRNAs and endocrine-disrupting chemicals related to preeclampsia have been identified, offering new perspectives for studying the occurrence and mechanisms of preeclampsia (PE). The content of the article is relatively complete and has a strong relevance to the theme.
At the same time, the following problems that puzzle me exist:
Question 1:In different literatures in the article, the assessment systems for the risk of chemical substances on PE are different. Will these differences have an impact on the comprehensive result analysis?
Response 1:
Thank you for raising this important point. Indeed, the use of different assessment systems for evaluating the risk of chemical substances on preeclampsia across various studies may contribute to variability in the comprehensive analysis. These differences can influence the interpretation of results, particularly when comparing findings across studies with diverse methodologies and criteria. The methodology varies based on the source of the sample used for analysis and the chemical supposed to be assessed. To address this, we have carefully analyzed and discussed the methodologies used in each study included in the review, highlighting their strengths and limitations. Additionally, where possible, we have focused on synthesizing common findings while accounting for methodological variability in our interpretations.
In future studies, establishing standardized assessment systems for chemical risks in PE would greatly enhance comparability and improve the robustness of comprehensive analyses. We will emphasize this need in the discussion section to provide more context for interpreting the findings.
Question 2:Some of the contents in Table 2 in the text have underlines. Are they for emphasis? This is what puzzles me.
Response 2:
Thank you for bringing this to our attention. The underlines in Table 2 were not intended for emphasis and were included inadvertently. We will remove them in the revised version of the manuscript to ensure clarity and consistency.
Question 3:The article points out that prenatal exposure to EDCs may increase the risk of PE by affecting epigenetics. Based on your literature review, what are the guiding suggestions for maternal protection and clinical practice?
Response 3:
Thank you for your insightful comment. Based on our literature review, we recommend implementing strategies to minimize maternal exposure to endocrine-disrupting chemicals (EDCs) during pregnancy. This includes reducing the use and frequency of products containing EDCs, such as certain plastics, cosmetics, and household chemicals, through informed consumer choices and education. Raising public awareness about the potential health risks associated with EDC exposure is critical to empowering individuals to make safer lifestyle decisions.
From a broader perspective, the establishment and enforcement of stronger regulations and policies governing the production, labeling, and permissible levels of EDCs in consumer products are essential. These measures would ensure reduced environmental and occupational exposure, providing an additional layer of protection for pregnant women.
Furthermore, integrating EDC risk assessment into clinical guidelines for prenatal care could help healthcare providers identify and advise high-risk populations. Together, these approaches would contribute to reducing the risk of preeclampsia and improving maternal and fetal health outcomes.

Reviewer 3 Report
Comments and Suggestions for Authors
In this well written systematic review the authors collate the peer reviewed literature linking in-utero endocrine-disrupting chemicals (EDC) exposure and microRNAs and their imprinted genes from prenatal and in maternal circulation and the placenta of PE.
All parameters of a full systematic review have been followed in a step wise manner and are well documented in the paper describing the search, the literature review and the assessment of bias in tables.
The authors utilized the data from 29 studies. They report that altered expression of 21 microRNAs (miR-15a-5p, miR-142-3p, and miR-185) in the placenta of PE patients that was positively associated with urinary concentration of phthalates and phenols in the first trimester. Furthermore they report a number of studies showing that phenols, phthalates, perfluoroalkyl substances (PFOA), 24 polybrominated diphenyl ethers (PBDE), and organochlorine phosphates (OCPs) have been associated with changes in hormones, growth factors and signalling pathways that have been linked with PE and hypertensive disorders in pregnancy. They also report on papers showing that miRNA-31, miRNA-144, miRNA- 26 145, miRNA-210, C14 MC, and C19 MC clusters may be used as possible targets for PE because of their potential roles in the onset and progression of PE.
This is a novel systematic review of an important topic. It was an interesting read, congratulations to the authors
Author Response
Reviewer 3.
In this well written systematic review the authors collate the peer reviewed literature linking in-utero endocrine-disrupting chemicals (EDC) exposure and microRNAs and their imprinted genes from prenatal and in maternal circulation and the placenta of PE.
All parameters of a full systematic review have been followed in a step wise manner and are well documented in the paper describing the search, the literature review and the assessment of bias in tables.
The authors utilized the data from 29 studies. They report that altered expression of 21 microRNAs (miR-15a-5p, miR-142-3p, and miR-185) in the placenta of PE patients that was positively associated with urinary concentration of phthalates and phenols in the first trimester. Furthermore they report a number of studies showing that phenols, phthalates, perfluoroalkyl substances (PFOA), 24 polybrominated diphenyl ethers (PBDE), and organochlorine phosphates (OCPs) have been associated with changes in hormones, growth factors and signalling pathways that have been linked with PE and hypertensive disorders in pregnancy. They also report on papers showing that miRNA-31, miRNA-144, miRNA- 26 145, miRNA-210, C14 MC, and C19 MC clusters may be used as possible targets for PE because of their potential roles in the onset and progression of PE.
This is a novel systematic review of an important topic. It was an interesting read, congratulations to the authors
Response:
Response:
We sincerely thank the reviewer for their thoughtful and encouraging feedback on our systematic review. We are delighted that you found the manuscript well-written and appreciated our efforts in following a rigorous systematic review methodology, including the stepwise approach, detailed documentation of the search process, and the assessment of bias.
Thank you for recognizing the importance of our research on this critical topic. Your supportive comments motivate us to continue exploring the intersection of EDCs, epigenetics, and pregnancy complications to further advance understanding and inform future clinical applications.
If you have any additional suggestions or areas where you feel the manuscript could be improved, we would be happy to incorporate your input.

Round 2
Reviewer 1 Report
Comments and Suggestions for Authors
The revisions are appropriate and sufficient.